# Slope Failure of Shilu Metal Mine Transition from Open-Pit to Underground Mining under Excavation Disturbance

Kang Yuan [1], Chi Ma [2], Guolong Guo [2] and Peitao Wang [2,*] 

[1] Hainan Mining Co., Ltd., Changjiang 610059, China; yuankang1000@163.com
[2] School of Civil and Resources Engineering, University of Science and Technology Beijing, Beijing 100083, China; machi@xs.ustb.com (C.M.); ggl06252012@outlook.com (G.G.)

\* Correspondence: wangpeitao@ustb.edu.cn

**Abstract:** The instability of slopes and ground subsidence caused by the conversion from open-pit to underground mining are important aspects of mining disaster research. This study focuses on the instability of slopes and ground subsidence during the conversion from open-pit to underground mining in the Beiyi mining area of the Shilu iron ore mine. Using numerical simulation and analysis, this study establishes a mechanical analysis model to assess the rock stability and movement of rockfall. The research findings indicate that there are significant stress concentration phenomena in the surrounding and floor areas of the goaf during the mining process. The collapse zone mainly develops in the western area before and after a certain level of mining and then shifts to the eastern part of Beiyi area. Surface subsidence expands after mining at a certain level, resulting in a large-scale disturbance area. Furthermore, the eastern slope experiences extensive landslides. This study suggests the continued monitoring of landslides and slope stability in specific areas of the mine. The research results can help us to understand the stability of the open-pit to underground rock mass in Hainan, judge the development trend of the surface subsidence range, and provide a reference for the stability evaluation of the rock mass mined by the open-pit-to-underground caving method.

**Keywords:** transition from open to underground; rock mass stability; the transport of collapsed bodies; numerical simulation

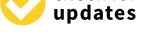



## 1. Introduction

Open-pit to underground mining is a common and complex mining system engineering project, in which the rock mass stability issues caused by combined mining disturbances are a hot topic of concern for many scholars [1–3]. The mechanical characteristics of rock masses under the disturbances of open-pit to underground mining are much more complicated than those of single open-pit or underground mining. The stability problems of rock masses, such as slope instability and mining subsidence, also exhibit significant differences compared to those under single mining conditions [4–7]. Shi et al. [8] took the Anjialing Coal Mine in Shanxi Province as an example to test the physical and mechanical properties of coal and rock in the laboratory. The similarity criterion is used to establish a similar experimental model of slope deformation evolution in an open-pit mine section, and the digital scattering method is used to test the influence of underground mining process parameters on the deformation evolution of an open-pit mine slope. Nguyen, P.M.V. [9] described a numerical analysis of slope stability in the Cao Son outcrop mine in Vietnam, where all calculated variables were assessed using the finite difference method code FLAC. To assess the stability of the OP slope, various geometrical configurations showing progress in OP and UG extraction were examined. Joyce Chung et al. [10] proposed a mixed integer programming model to determine the optimal transition point and transition period from open-pit to underground mining, maximizing the net present value of the project, while considering the roof column arrangement and development costs. E. Bakhtavar [11] proposed a model based on the economic value of blocks based on open-pit and underground

methods, as well as the net present value (NPV) obtained through mining. In the model, the NPV of the open-pit mine is compared with that of the underground mine to determine the optimal transition depth. Hongge Peng et al. [12] established a thin plate model of a mining slope based on the reality of combined mining and underground mining and its interaction, analyzed the influence of underground mining on the slope, and obtained the deformation law of a mining slope under open-pit and underground mining conditions from the change in rock layer dip caused by roof deformation. Li et al. [13] conducted similar simulation experiments and used digital measurement techniques to analyze the deformation characteristics of surrounding rock using the room-and-pillar method during the open-pit-to-underground transition. This study found that the surrounding rock underwent three stages: localized micro-movement, linear large deformation, and overall nonlinear collapse after mining. Meanwhile, with an increase in the dip angle of the phosphate ore body, the failure angle decreased, and, at 50°, the surrounding rock exhibited a feature of overall shear failure. Wang et al. [14] conducted FLAC3D numerical simulations and similar physical model experiments to analyze the displacement and stress changes and the distribution of plastic zones in the composite mining system. They obtained a linear correlation between the displacement changes of overlying rock layers in the mining area and the variation in the mining-induced influence zone and excavation space. Feng et al. [15] studied the failure characteristics of overlying rock masses during full extraction of the open-pit to an underground L-shaped panel in the Wuhai open-pit coal mine using numerical simulation methods. Du et al. [16] studied the slope stability before the transition to underground mining using limit equilibrium theory and FLAC3D v5.0. They analyzed the stress variation, distribution of plastic zones, and displacement changes under different mining levels and conditions with and without backfilling in the subsidence zone, defining the surface deformation as the criterion for delineating the surface movement range at different mining levels. Jaroslav et al. [17] analyzed the stability changes in the slope rock mass structure after the transition from open-pit to underground mining in the Ekati diamond mine from a geological survey perspective. Zhao et al. [18] established a three-dimensional finite difference numerical model of the Jinfeng gold mine using FLAC3D, including the surface topography, ore body, shafts, and major faults. Based on this, they analyzed the shaft stability and surface deformation caused by the transition from open-pit to underground mining. Dintwe et al. [19] conducted numerical simulations using FLAC3D software to investigate the behavior mechanism of roof pillars during the underground mining process in the transition from open-pit to underground mining, and analyzed the stress distribution and failure mechanism around the roof pillars. Wang et al. [20] used the structural orientation analysis software DIPS version 5.01 to statistically analyze the structural orientation of siliceous limestone and tuffaceous limestone and provided a detailed analysis of the formation and development mechanisms of surface subsidence. Wang et al. [21] conducted research on the failure forms of surrounding rocks from a microscopic perspective based on the Particle Flow Code (PFC) software version 3.1, revealing the intrinsic laws and particle movement patterns of slope failure. Zhao et al. [22] established a surface rock movement monitoring network using GPS, obtained displacement changes, and then utilized three-dimensional numerical simulation methods such as FLAC3D to investigate the mechanical environment of the surrounding rocks and slopes on open-pit slopes during underground mining. They explored the deformation of the surrounding rocks, stress field distribution, and failure mechanisms. Wang et al. [23] employed a combination of similarity experiments and numerical simulations to study the failure characteristics of slopes in the transition from open-pit to underground mining and their impact on underground ore extraction. They analyzed the damage and failure characteristics of slope rock mass at different mining levels. From the above studies, it can be seen that for the large-scale and complex engineering problems of the open-pit-to-underground transition, numerical simulation methods are mainly used to study the overall stability of the project, while the continuum mechanical analysis method is mostly used in the study of the overall stability of the project, and the overburden rock layer above the excavated ore body is simplified to a continuous rock layer

for calculation [24–26]. The aforementioned studies are of great significance for elucidating the deformation mechanism, mechanism analysis, and stability research of rock masses in the transition from open-pit to underground mining, and they provide a reference for the stability evaluation of the rock mass mined by the open-pit-to-underground caving method.

To describe the geomechanical phenomena that occur under the conditions of underground exploitation, this paper deals with the problem of open-pit slope stability and ground surface under the influence of underground mining of the Shilu iron deposit in China. Simulations with the help of numerical computational models, using the FLAC3D code, were conducted to study the influence of staged exploitation of the ore body on the slope's stability. This work may provide a methodology reference for the analysis of rock stability in open-pit to underground mines.

## 2. Overview of the Beiyi of Shilu Iron Mine

### 2.1. Geographic Location and Geohydrological Environment

Shilu Iron Mine, owned by Hainan Mining Co., Ltd., is located in Shilu Town, Changjiang County, western Hainan Province, China. The mining area is situated in the northwest branch of Wuzhi Mountain and belongs to a low mountainous terrain. The eastern, southern, and western sides are characterized by medium-to-low mountains, mainly composed of slate and quartzite rocks. The northern part consists of undulating granite lowlands, with the terrain gradually sloping from south to north, as shown in Figure 1.

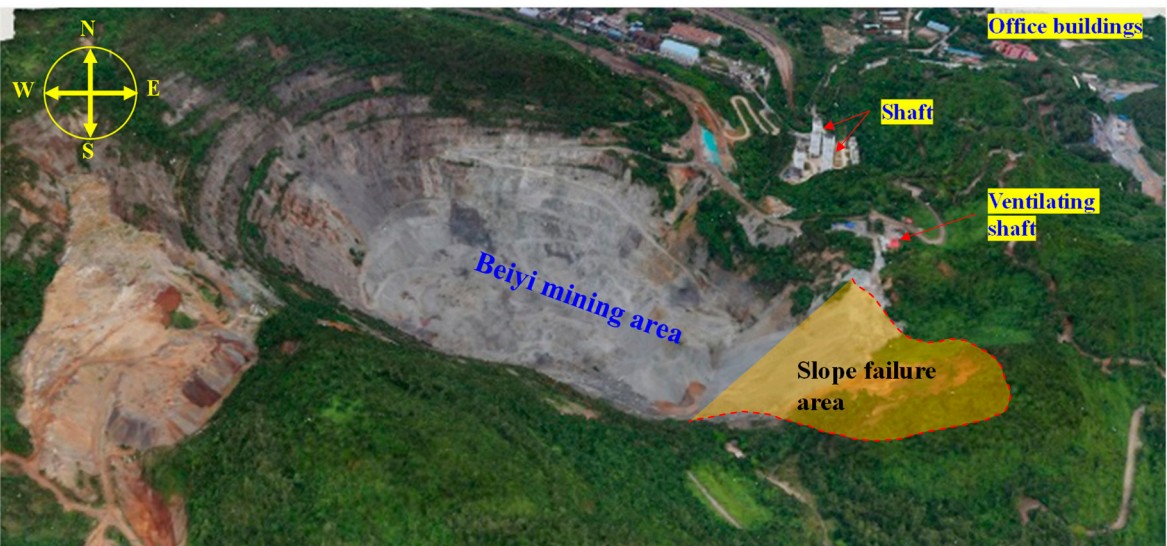

**Figure 1.** Topography of Beiyi mining area.

The iron, cobalt, and copper ore bodies in the Shilu iron ore mine area are sur-rounded by hard state rocks, mainly dolomite, tremolite, and silicified dolomite, with a local distribution of schist. The geological conditions of the mine area are simple, and the geological exploration type is Type III (medium type of laminated rocks). There are several faults in the tectonic mine area, but, in the Beiyi mining area, the faults have little influence on the overall rock stability. The hydrogeological conditions of Beiyi quarry are of the simple type; the groundwater is mainly weathered fracture water and tectonic fracture water, and the dynamic and static reserves are not abundant, and so the water volume changes are directly related to atmospheric rainfall, and there is no connection with surface water. The main water-bearing layers are the seventh and sixth layers of the Shilu Group, and the fifth layer is the water barrier of the mine area, with poor overall permeability. The water filling type of the deposit is mostly tectonic fissure filling, which is not influenced by the surface water body.

## 2.2. Rock Mass Properties

The surrounding rocks above and below the iron, cobalt, and copper ore bodies in the Shilu iron ore mine area are mainly hard state rocks, mainly dolomite, diorite, and silicified dolomite, and schist is only locally distributed. The stability of the ore bodies and their surrounding rocks above and below is generally good. However, the development of folding and fracturing in this area has led to the fracturing and modification of rocks along the fractures and faults and to the internal extrusion of fissures. Consequently, the strength of the rocks is significantly reduced. According to an analysis of engineering properties and the distribution characteristics of geotechnical bodies in various sections of the mine area, the geological conditions of the mine area are of a simple type, and the geological exploration type is Type III (medium type of laminated rocks). In addition, several faults of a relatively large size occur in the tectonic mine area, but, in the area studied, the fault runs through the center of the Beiyi quarry. The faults may have some influence on the movement of the rocks locally, but they will not control the rock movement pattern as a whole. Hainan Shilu iron ore involves mainly steep-dipping joints, and slow-dipping joints are not developed.

In order to obtain accurate rock mechanical parameters, the physical and mechanical indexes of each rock in the underground and mining areas were obtained through multiple processes such as on-site sampling, the preparation of specimens, a density test, wave velocity test, mechanical experiment, and data analysis, and the mechanical parameters of rock mass were determined by combining the Hoek and Brown empirical speculation methods. The density of Shilu iron ore schist is about 2798.37 kg/m$^3$, and the wave velocity is 5337.88 m/s. The density of the double permeable rock is about 2899.95 kg/m$^3$, and the wave velocity is 5759.31 m/s. The hematite is about 3675.58 kg/m$^3$ and the wave velocity is 5781.59 m/s. The average uniaxial compressive strength of schist in the North No. 1 mining area is 72.04 MPa, the elastic modulus is 29.55 GPa, the average secant modulus is 16.55 Gpa, and the Poisson's ratio is about 0.35. The uniaxial compressive strengths of bipermeable rock and hematite are 68.09 MPa and 106.44 MPa, the elastic moduli are 36.97 MPa and 65.12 MPa, and the Poisson's ratio is 0.33. The average cohesion of the joint plane of the rock mass was 0.323 MPa, the average friction angle was 37.965°, and the average friction coefficient was 0.786. This provides important mechanical data for the subsequent mechanical analysis of engineering rock mass and the numerical simulation of mining subsidence.

## 2.3. Rock Movement Problem

The slope height of the open-pit mine of Shilu iron mine is 12 m, the width of the slope design step is 6–10 m, the slope angle of the open-pit mine is 60–70°, and the final slope foot is 29–49°. It was transferred to underground mining in June 2018, with a single underground mining height of 15 m, and the mining of ore bodies above −45 m was completed, primarily using the bottomless column segmental caved method. The ore body to be mined underground in the Beiyi mining area is located below the bottom of the open pit and its east gang, and it continues to extend to the southeast of the open pit. Therefore, the disturbance of the ore body mining has a great impact on the slope rock of the bottom of the open pit and its southeast range. The mining span of the caved mining area is large, and the interior is filled with the overburden rock formed by the caved surrounding rock. The overburden rock in its bulk state has a low shear strength and is extremely deformed under pressure, and it has almost no support for the overburden rock layer. After mining, the load and self-weight of the top slab are supported by the top of the empty area and the side gang rock. Under the action of the self-weight of the rock body and the tectonic stress, the rock body undergoes overall shear damage and the top slab and side gang rock collapse into the empty area, which will lead to strong rock destabilization and mining subsidence problems. This effect will be more serious in a deep concave open-pit terrain, and will also cause landslide problems on the slope of the quarry or the original natural slope. In addition, a 40 m thick overburden was reserved at the beginning of the mine, and,

after years of mining, the overburden thickness is now tentatively judged to be over 60 m. Such avalanche transport will significantly affect the stability of the rock mass. Therefore, the mine should fully consider the impact of caved body transport, understand the stability of the open to underground rock mass as soon as possible, and judge the development trend of the surface collapse range, which is of great significance to the safe management and efficient production of the mine.

## 3. Numerical Model

### 3.1. Simplification Principle of Model Calculation

According to the field investigation, the lithology of the mining area is very complex, and the distribution of rocks and minerals is almost irregular. In terms of surrounding rock, because the ore body is deposited in the Shilu Group, and because the group is divided into three layers, the main rocks are double-permeable rock and double-permeable dolomite, which have low strength and poor water resistance. So, the mechanical properties of the rock disturbed by underground mining are taken from the rock. In terms of ore, it is difficult to determine the three-dimensional morphology of the ore body based on the profile, so the ore body is built into a grid shape based on the approach. Outside the outlying layer of the mining area, the gently dipping joints are not developed, and the dip angles of most joints are greater than 65°. Under the cutting of steep dipping joints, the layer is conducive to the generation of a falling failure mode. Since the tensile strength of the rock is low enough and the influence of the joints throughout the surrounding rock is taken into account regarding the mechanical parameters of the surrounding rock, the influence of the joint surfaces is not considered separately in the model. F25 and F31 faults are developed in the mining area, but the faults pass through the central position of Beiyi and Baoxiu stopes. These have a certain impact on the migration and fall of the surrounding rock but do not control the boundary of the subsidence area, so in order to reduce the complexity of calculation, the influence of faults on the movement law of rock strata is not considered.

According to the on-site sampling, the specimens were prepared, the rock mechanics experiments were carried out, the rock mechanical parameters were obtained, and the rock mass mechanical parameters were determined, as shown in Table 1.

**Table 1.** Mechanical parameters of rock mass.

| Modulus of Elasticity | Poisson's Ratio | Internal Friction Angle | Compressive Strength | Tensile Strength | Cohesion |
|---|---|---|---|---|---|
| 65.12 MPa | 0.3 | 35° | 106.44 MPa | 8.87 MPa | 25 kPa |

### 3.2. Principle of Numerical Simulation Method

Considering the transport of a caved body, the caving method is used to control the ground pressure through collapse, and the damage form of typical caved mining surrounding rock is shown in Figure 2. The simulation should not only consider the subduing, softening, and failure of the surrounding rock area but should also consider the impact of the surrounding rock on an expansive bulk filling goaf after failure. Additionally, it should have a supporting effect on the surrounding rock in the empty area, that is, the influence of caving body migration.

Therefore, under the condition of considering the influence of caved body transport, when performing rock stability simulation analysis, instead of using large deformation calculation, the caved area is considered as Eulerian description; the Mohr–Coulomb ideal plasticity criterion is calculated for the caved body; and the strain-softening criterion is calculated for the surrounding rock around the caved body.

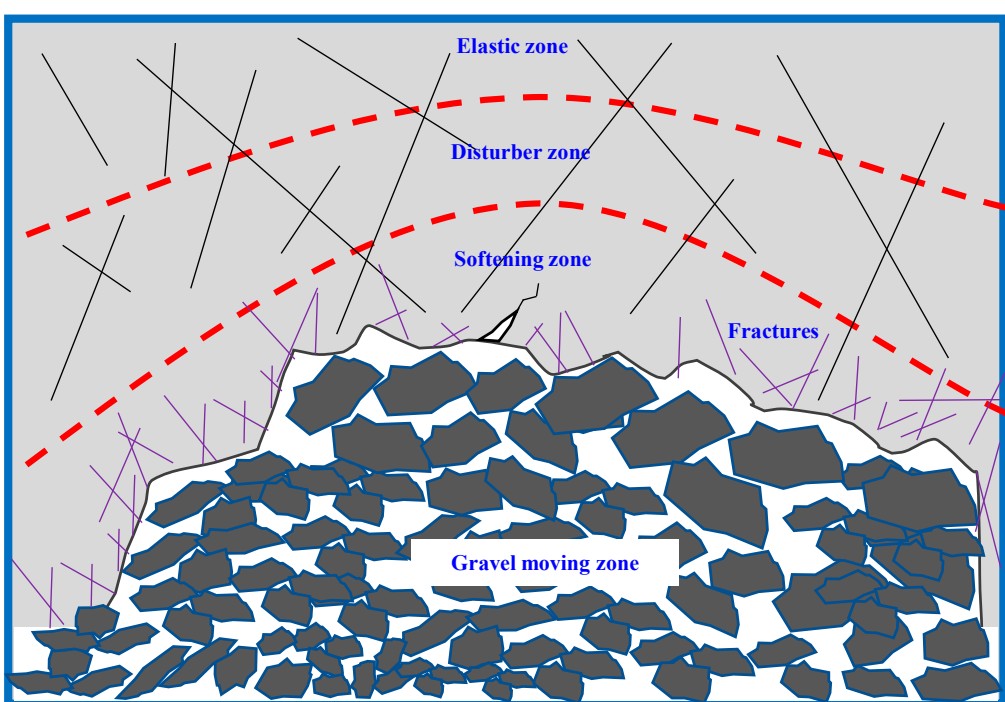

**Figure 2.** Schematic diagram of caving in caving mining.

(1) The Mohr–Coulomb criterion is implemented using the following steps:

1. The elastic principal structure is used to calculate $\sigma_x$, $\sigma_y$, and $\sigma_{xy}$ for the new step. The principal stresses $\sigma_1$ and $\sigma_3$ and their directions are calculated. $\sigma_1 < \sigma_3$, i.e., $\sigma_1 < \sigma_3$ is the maximum principal stress (if the pull is positive).

2. The shear and tensile damage criteria for rocks are shown in Equations (1) and (2), respectively, and $\sigma_1$ and $\sigma_3$ are substituted to determine whether damage occurs.

$$f_s = \sigma_1 - \sigma_3 N_\varphi + 2c\sqrt{N_\varphi} \tag{1}$$

$$f_t = \sigma_3 - \sigma_t \rangle 0 \tag{2}$$

$$N_\varphi = \frac{1 + \sin\varphi}{1 - \sin\varphi} \tag{3}$$

where c is the cohesive force; $\varphi$ is the angle of internal friction; and $\sigma_t$ is the tensile strength.

3. If shear damage occurs, then the principal stress is recalculated according to Equations (4) and (5) and

$$\sigma_1^N = \sigma_1 - \lambda_s(a_1 - a_2 N_\psi) \tag{4}$$

$$\sigma_3^N = \sigma_3 - \lambda_s(a_2 - a_1 N_\psi) \tag{5}$$

Among them,

$$\lambda_s = \frac{f_s(\sigma_1 + \sigma_3)}{a_1 - a_2 N_\psi - (a_2 - a_1 N_\psi)N_\psi} \tag{6}$$

$$a_1 = K + \frac{3}{4}\sigma \tag{7}$$

$$a_2 = K - \frac{2}{3}\sigma \tag{8}$$

$$N_\psi = \frac{1 + \sin\psi}{1 - \sin\psi} \tag{9}$$

where $\psi$ is the shear expansion angle, and for the associative flow law, we have $\psi = \varphi$.

4. If tensile damage occurs, the principal stresses are recalculated according to Equations (10) and (11).

$$\sigma_1^N = \sigma_1 - \lambda_t a_2 \tag{10}$$

$$\sigma_3^N = \sigma_3 - \lambda_t a_1 \tag{11}$$

Among them,

$$\lambda_t = \frac{f_t \sigma_3}{\sigma_1} \tag{12}$$

5. $\sigma_x$, $\sigma_y$ and $\sigma_{xy}$ are recalculated based on the maximum and minimum principal stresses as well as the principal stress directions.

(2) The strain-softening criterion, i.e., the strength of the rock (body) material, weakens with increasing plastic strain; therefore, strain softening can be achieved based on the Mohr–Coulomb ideal plasticity criterion by weakening the values of c and $\varphi$ according to the equivalent plastic strain.

The avalanche bulk medium is transported from one cell to another according to the nodal velocity of the Eulerian mesh. The transport of the collapsed body satisfies the mass conservation equation, i.e.,

$$\frac{\partial \rho}{\partial t} + v_i \rho_{,i} + \rho v_{i,i} = \frac{\partial \rho}{\partial t} + (\rho v_i)_{,i} = 0 \tag{13}$$

In the above equation, $\rho$ is the density of the avalanche and $v$ is the velocity.

Considering each cell as a control volume, according to the Gaussian dispersion theorem, the density of each control volume varies as

$$\int_v ((\rho v_i)_{,i}) dv = \int_s \rho v_i n_i d_s = \Sigma_f \overline{p}^{(f)} \overline{v}_i^{(f)} \overline{n}^{(f)} \tag{14}$$

In the above equation, the superscript (f) indicates the face of the control volume.

The conversion relationship between the voidness $\eta$ of the caved body and the density $\rho_d$ is

$$\rho_d = \rho_s (1 - \eta) \tag{15}$$

In the above equation, $\rho_s$ the particle density of the caved bulk can be taken as the density of the original rock, and the bulk composed of a variety of materials can be taken as the average.

(3) Arithmetic process.

When using FLAC 3D to simulate the transport of caved media, the whole flow process cannot be simulated very accurately because momentum conservation is not considered and local damping is used. However, for the caved problem of caved mining, what is more critical is the caved body transport needed to fill the empty area; therefore, numerical simulation can still reflect this core problem despite certain defects. Because of the Eulerian description of the debris unit, both the rock unit and the debris unit are considered in the model. The rock body in the rock unit is transformed into the rubble unit when the equivalent plastic strain reaches a critical value. In addition, during the calculation, the relevant intrinsic parameters are continuously updated according to the porosity of the units. The calculation process is shown in Figure 3.

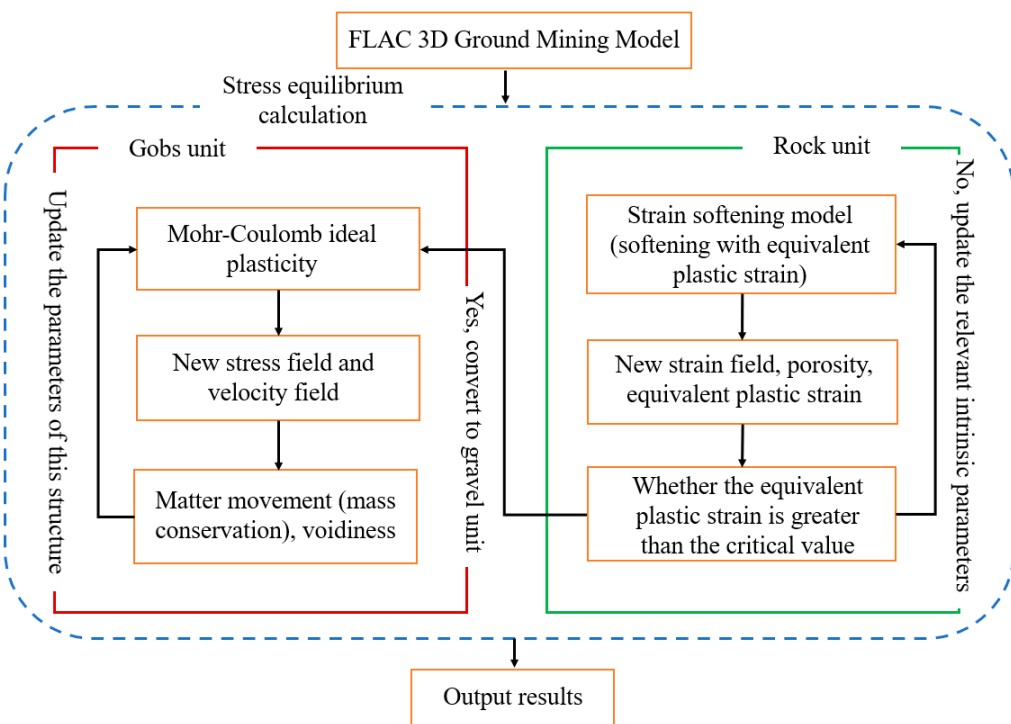

**Figure 3.** The variation in the elastic modulus of collapsed body and original rock with porosity.

*3.3. Model and Calculation Process*

The stability analysis model of the Beiyi quarry consists of overburden rock, ore body, softening zone rock, plastic zone rock, and peripheral geological rock. The overall length is 4536 m, the width is 3528 m, and the elevation of the model base is −720 m. The numbers of grids and nodes in this model are 814,812 and 900,478, respectively, and the specific model and ore body are shown in Figures 4 and 5. Table 2 shows the assignment of mechanical parameters.

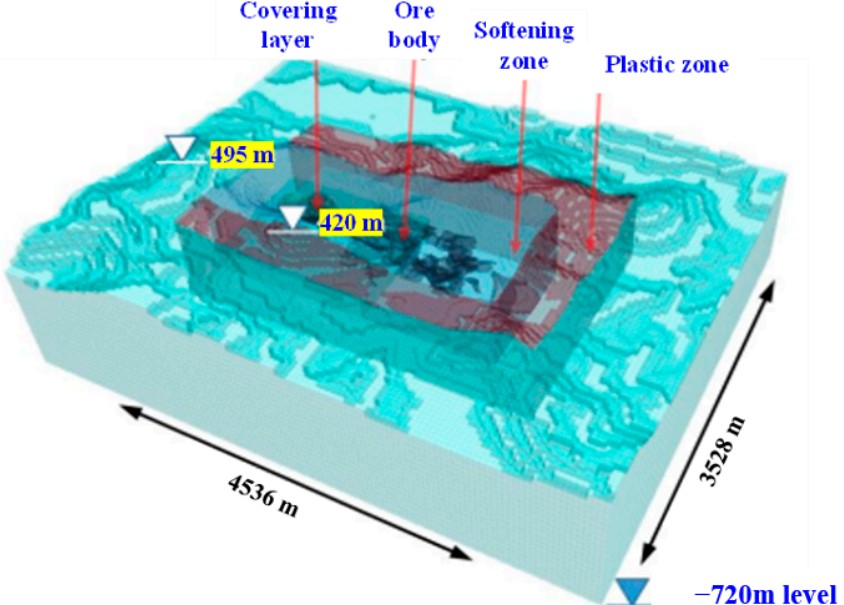

**Figure 4.** The 3D numerical calculation model.

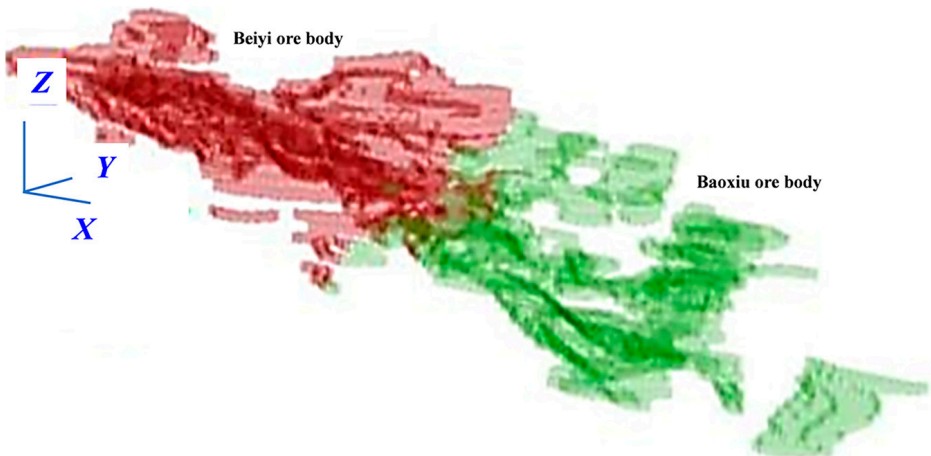

**Figure 5.** Beiyi ore body (red indicates west of E9 and green indicates east of E9).

**Table 2.** Model mechanical parameters.

| Lithology | Modulus of Elasticity | Poisson's Ratio | Internal Friction Angle | Compressive Strength | Cohesion |
|---|---|---|---|---|---|
| Hematite | 65.12 MPa | 0.33 | 57.91° | 106.44 MPa | 3.25 kPa |
| Surrounding rock | 36.97 MPa | 0.26 | 42.55° | 68.09 MPa | 1.62 kPa |

The ore body model is "stacked" according to the distribution of the mining approach. From the distribution of the ore body and surface topography, the main iron ore body in the north extends in a southeasterly direction, with an overall orientation of 315~135° and an assigned elevation of 0~−602 m. The preliminary design mining range is 0~−360 m. The morphology is generally stratified-like-stratified, and it is continuously distributed along the strike direction. As can be seen from Figure 6, the overall distribution of the ore body mining range is bounded by E9, with the west ore body west of E9 located below the bottom of the open pit, and the east ore body east of E9 located below the east gang of the open pit and its eastern hills. The simulated mining area is divided into six stages: +30~0 m, 0~60 m, 60~120 m, 120~180 m, 180~240 m, and 240~360 m.

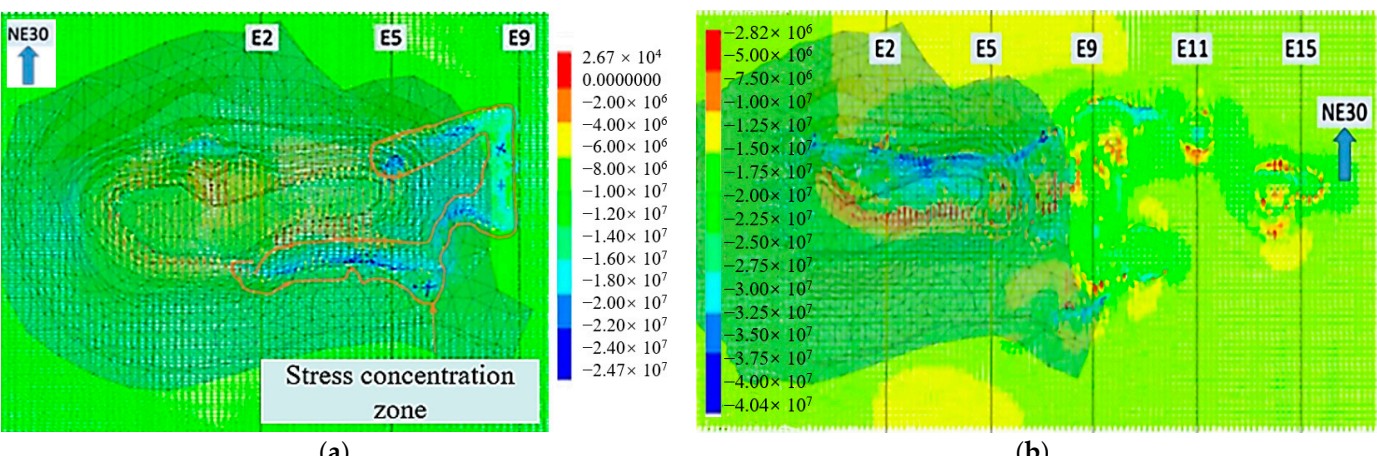

**Figure 6.** Distribution characteristics of stress field around goaf at different levels after mining in Beiyi mining area (maximum principal stress, unit Pa). (**a**) is the 0 m level and (**b**) is the −120 m level.

After completing the above model, the calculation process is briefly described as follows: assigning parameters to the model and imposing boundary conditions, imposing constraint boundary conditions on the sides of the whole model, and setting fixed boundary conditions at the bottom; making the model reach static equilibrium under the action of gravity, and then applying ground stress (see the next section for the specific content and effect of ground stress application), calculating the initial equilibrium calculation after equilibrium, zeroing the obtained displacement, and simulating the basic state of the ore rock before mining; opening the caved calculation mode, setting the calculation parameters related to the caved simulation; taking each stage as a step for mining calculation; calculating the changes in the surrounding rock stress field and the corresponding surrounding rock deformation after the end of segment mining until the calculation process stabilizes; and repeating the above mining steps until the end of mining of the ore body in the whole mining area.

## 4. Stability Analysis

### 4.1. Stress Distribution Characteristics of the Surrounding Rock

The stress distribution of the surrounding rock in the retrieval area of the Beiyi quarry mined to the 0 m and −120 m levels is shown in Figure 6. In order to clearly reveal the state of stress distribution, only the stress vector field within 30 m above and below the mined level is given in the figure, and the color of each stress indicates the maximum principal stress value.

When mining to the 0 m level, the stress concentration area is distributed in a circular pattern along the recovery area, mainly existing in the foot of the slope of the side gang of the open pit, the pit bottom area, and the side gang rock bodies of the north gang and the east gang. Among them, the stress concentration area where the maximum principal stress reaches 20 MPa is mainly concentrated in the north gang and east gang hill, ranging between E2 and E9 profiles, and the level of the stress concentration zone is about 30–40 m wide. The maximum value of the maximum principal stress is 24.7 MPa, and the return void at this location is narrow (the wide range of the return area cannot easily produce a stress concentration phenomenon). The overlying rock layer thickness at this location is large, the side slopes are high, and a large range of hollow surfaces exist in the rock bodies of the north gang and east gang involved. In addition, there is also a certain amount of stress concentration in the bottom plate of the mining area, but the overall amount is less than 20 MPa.

When mining to a −120 m level, according to the test results of ground stress, the maximum principal stress at this level is about 18 MPa. The main area of stress concentration is located on the north side of the recovery area, the maximum value is about 40 MPa, and the stress concentration level of the bottom slab is not more than 30 MPa. It can be seen that the overall value is still not high when mining to this level.

When mining to the −240 m level, at this time, west of E9 (i.e., Beiyi west area), mining has been completed, and the stress distribution is shown in Figure 7. This level includes two retrieval areas in the south and north, and the stress concentration area is also mainly present in the surrounding area of E9 and E11 profiles in the two retrieval areas. In these areas, the stress concentration level in the southern mining area is higher, and the maximum amount reaches 48 MPa, while there is no obvious stress concentration phenomenon on the north side.

### 4.2. Analysis of the Development Morphology of the Caved Zone

Based on the numerical calculation results, the morphology of the development of the caved zone is analyzed in conjunction with the orebody fugacity. The morphology of the cave in zone was fitted with a smooth surface. The morphology of the caved zone in the Beiyi quarry mined to −60 m, −120 m, −180 m, −240 m, and −360 m is shown in Figure 8.

There is a mining site north with −60 m level mining and a caved area mainly in the area west of E9 development. This is because the ore body west of E9 is lower, the −60 m

ore body is mostly in the west mining area range, open-pit bottom area, and the southeast end of the gang below the ore rock cave. In the E9 profile, near the boundary, the bubbling area is not developed to reach the surface, as seen in Figure 8a. The lower part of the side gang rock cave will cause the foot of the slope sink and the upper part of the rock fissure to sprout and expand, which will lead to the problem of a landslide in the east gang of the open pit. With the mining to the −120 m level, the caved zone develops to the southeast, and then the caved zone in the E9 profile develops to the surface, as seen in Figure 8b. However, the general extent of the caved zone in the open pit did not experience a large change. The rock layer in the Beiyi quarry is a diagonal structure and the ore body below 0 m extends to the north, so the south gang of the open pit is represented by the lower plate. In addition, a 40 m thick cover layer is laid at the bottom of the open pit, and the ore body in this area has a large interlayer, which can be crumbled and filled with debris during the mining process, all of which can effectively prevent the development of an open pit. This can be seen from the direction of surface material movement at one point, as shown in Figure 9.

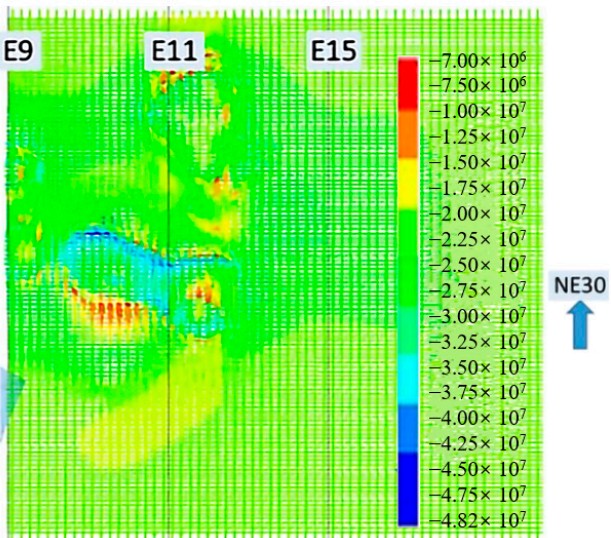

**Figure 7.** Distribution characteristics of stress field around the mined-out area at the level of −240 m in Beiyi mining area (maximum principal stress, unit: Pa).

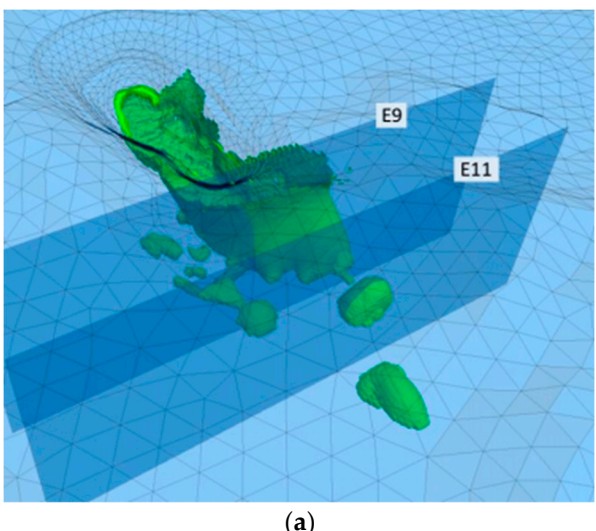

(**a**)

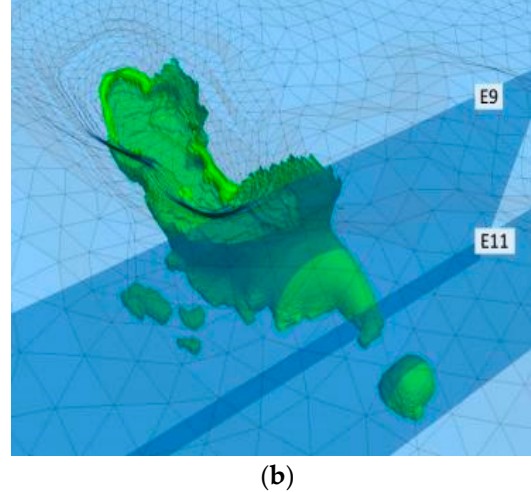

(**b**)

**Figure 8.** *Cont.*

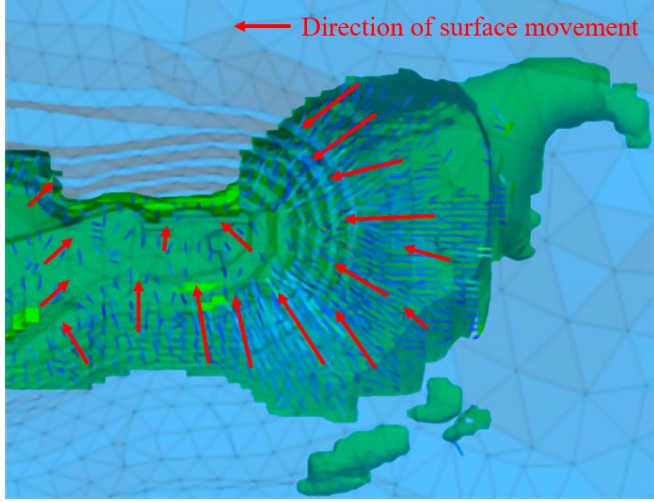

**Figure 8.** The northern mining area is mined to the boundary of the subsidence area at different levels (green represents the boundary of the subsidence area). (**a**) is the −60 m level, (**b**) is the −120 m level, (**c**) is the −180 m level, (**d**) is the −240 m level, and (**e**) is the −360 m level.

**Figure 9.** Direction of movement at a certain moment when mining to −60 m surface in Beiyi mining area.

From −120 m onwards, the ore body deposit in the Beiyi west zone decreases, while the ore volume in the Beiyi east zone gradually increases. At this time, the development of the caved zone is mainly located in the northeast zone. Since the ore body in the east zone is buried at a greater depth and is gently dipping, the caved zone gradually emerges upward. However, by the end of mining (−360 m), the caved zone did not develop to the surface either. By measuring the vertical distance between the boundary of the caved zone and the surface, it was found that at the end of mining, the top of the caved zone near the E11 profile was still 220 m away from the surface, and the moving mining void in the E13 profile was 539 m away from the surface. In addition, since mining, the caved zone has not developed towards the west gang of the open pit. Because the underground ore body is far away from the west gang, it is beneficial to the stability of this gang.

### 4.3. Analysis of the Movement of the Surrounding Rock

The surface displacements of the ore body in the Beiyi mining area at different stages of back mining are shown in Figure 10. The blue part of the figure is the part with displacement over 1 m, which can be regarded as the surface collapse zone, and the part with displacement over 0.1 m is regarded as the sinkhole zone, which is divided by color according to the specific magnitude; the part with a displacement less than 0.1 m is not shown. It can be seen that as the mining proceeds, the surface collapse zone gradually increases and the sinkhole range develops more rapidly.

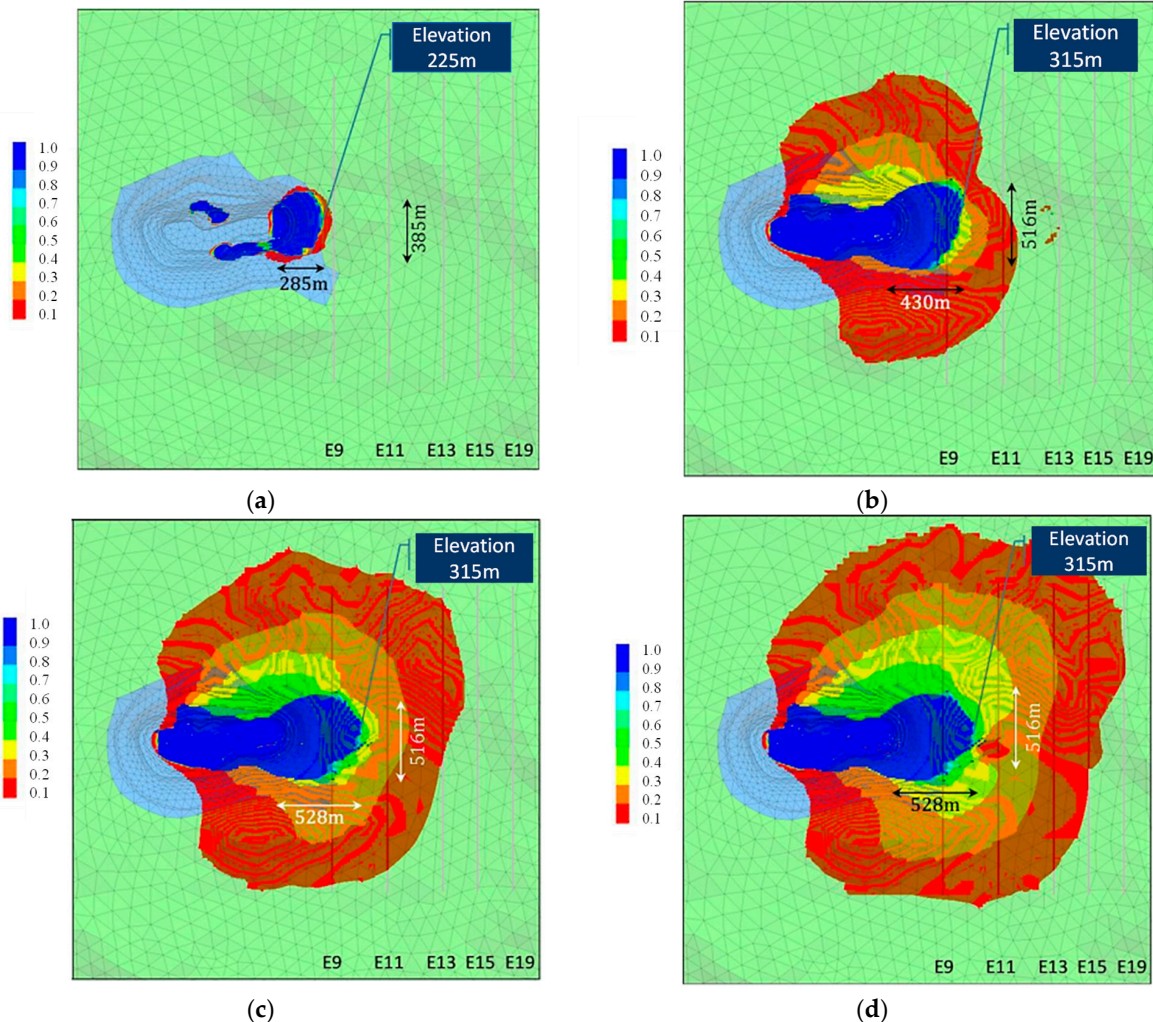

**Figure 10.** Surface displacement at different levels during mining in Beiyi mining area (unit m). (**a**) is the 0 m level, (**b**) is the −120 m level, (**c**) is the −240 m level, (**d**) is the −360 m level.

From Figure 10a, it can be seen that when mining from a +30 m to 0 m level, the collapse area of the three parts of the original side gang gradually increases, among which the collapse area of the east gang expands to the southeast to the upper elevation of 225 m. At this time, the range of surface displacement greater than 0.1 m was basically near the collapse area. At this time, the surface was not affected by a large range, and it was limited to the area around the east gang and the foot of the south and north gang slopes. After mining to a −120 m level, with a 0 m elevation at the bottom of the open pit as the reference, it can be seen that the size of the eastern caved zone in the plane increased from 285 m × 385 m to 430 m × 516 m. The sinkhole area obviously expanded to the east, south, and north, forming a wide range of mining disturbance area. Comparing Figure 10a,b, it can be seen that when the well was mined to −120 m, the east of the slope foot of the west gang of the open-pit side gang was a collapsed area on the east side, which was extended to 315 m.

From Figure 10c, it can be seen that the collapse zone to the west of E9 is basically unchanged when mining reaches −240 m, which is consistent with the orebody deposit. Compared with the 0 m mining level, the surface displacement is more than 0.1 m on both sides of the open pit at the −120 m level, which is mainly due to the maximum main stress unloading caused by the mining of the ore body. The stress unloading of the rock layer then produces a large horizontal displacement. Comparing the extent of surface cave-in zone, shown in Figure 10c,d, it can be seen that the extent of surface cave-in zone basically does not change from −240 m to −360 m at the mining level, which is mainly due to the small scale of the ore body below the −240 m level in the Beiyi East quarry, and it is difficult to develop the bubble fall of the surrounding rock on the surface.

## 5. Site Situation Investigation and Comparison

According to the site observation report, at the end of the 30 m level of mining in Beiyi quarry, a large range of landslides occurred on the surface side gang of the open pit. In order to further analyze the stability of the surface rock body of the open-pit to the underground area at that time, an open-pit-to-underground stability analysis model considering the transport of the caved body was used to simulate the mining of the ore body at a level of 30 m and above. In the model, rock units that exceeded the critical equivalent plastic strain were transformed into debris units, and units with a unit displacement exceeding 1 m could be considered as slides or cave-ins, and units with surface rock slides exceeding 1 m in the model were specially marked with colors, as shown in Figure 11. The area where the landslide occurred at the site is in the east gang, and a realistic model of the landslide area is taken at the site using the UAV tilt photography technology, as shown in Figure 11. It can be seen that when the Beiyi quarry was mined to a 30 m level, the east gang of the open pit was significantly damaged on a large scale, and the simulation results were very consistent with the actual situation on site. From the simulation results, combined with the change characteristics of the displacement field and caved area under different levels, under the disturbance of ore body mining at a −30 m level, the lower part of the east gang crumbled; the foot of the slope and the lower surrounding rocks were destroyed, starting from the foot of the slope; the rocks of the east gang adjacent surface collapsed to the bottom of the pit; and, gradually, a large-scale landslide phenomenon occurred. In the future, the mine will continue to disturb the rock body of the east gang when mining the horizontal ore body below −30 m. The mine should pay long-term attention to the problem of landslides in the east gang. In addition to the obvious landslides, the mine should also pay attention to the local stability of the central slope of the south gang and the north gang of the open pit, where the displacement of the foot of the central slope of the south gang is obvious, and landslides may occur under the subsequent mining disturbance.

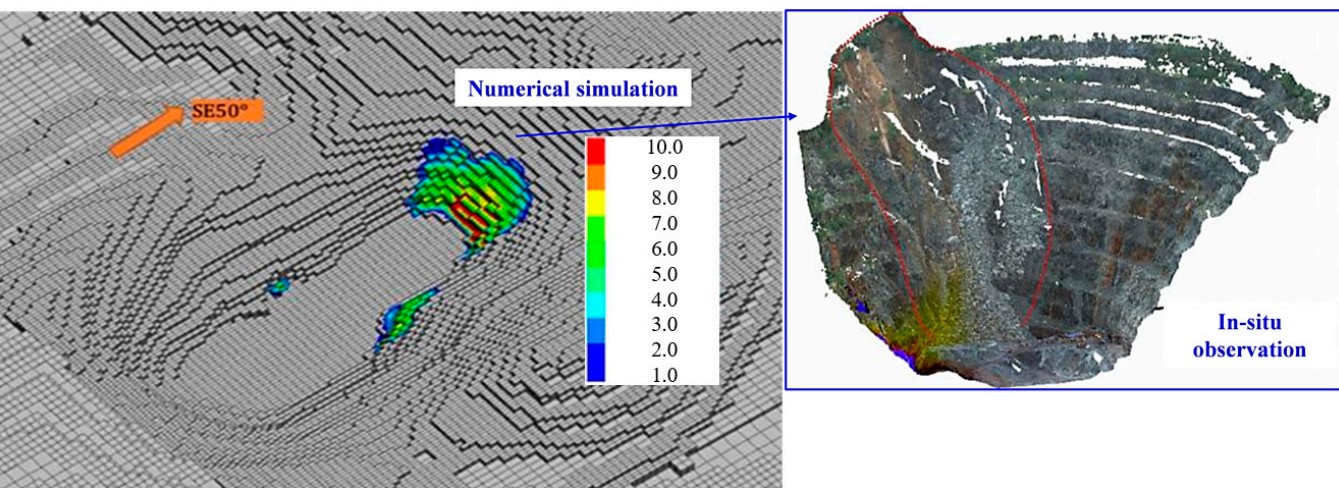

**Figure 11.** In situ observation of collapse of the eastern area of Beiyi mining area.

### 6. Conclusions

Based on the FLAC 3D numerical simulation platform, the Mohr–Coulomb plasticity criterion is used to simulate the caved body, the strain-softening criterion is used to simulate the affected rock mass, and a three-dimensional numerical analysis model is established to study the stability of rock mass during the open-pit-to-underground mining process based on the full consideration of caved body migration factors. This provides a methodological reference for the analysis of rock stability in open-pit to underground mines. The following conclusions were obtained:

(1) After mining at the 0 m level, the stress concentration in the rock mass around the recovery area is obvious, and a 30–40 m wide stress concentration zone is formed between the E2~E9 profiles of the north gang and the east gang, and the subsequent mining stress is mainly concentrated in the north side of the recovery area and gradually develops toward the E11 profile. When mining reaches the −240 m level, the stress is concentrated in the surrounding area of E9 and E11 profiles, and the stress concentration around the south mining area is significant.

(2) The development direction of the caved zone changes with the ore body's storage conditions, mainly to the east, with less influence on the stability of the west gang rock. After the end of mining at a −60 m level, the caved zone mainly develops in the area west of E9, and, from −120 m onwards, the caved zone is mainly located in Beiyi East. Under the influence of the distribution of the caved zone, the east gang of the open pit and the surrounding rock masses are displaced toward the bottom of the open pit, while the west gang rock masses are not affected by the caved zone. When mining from a +30 m to 0 m level, the range of surface subsidence zone is small, and after mining at a −120 m level, the range of subsidence increases rapidly, covering the whole pit bottom area and expanding to the east, south, and north, forming a large mining disturbance area.

(3) At the end of mining to the 30 m level in the Beiyi quarry, the mining disturbance caused the foot of the east gang slope to sink. The overall rock body was destabilized, resulting in a large-scale landslide toward the bottom of the open pit, and the simulation results matched the actual situation. In a future production process, the mine should also continue to pay attention to the development of landslides in the east gang of the open pit and the stability of the rock body of the central side slope of the south gang and the local side gang of the north gang.

**Author Contributions:** K.Y.: in situ investigation, data analysis, writing—original draft, and writing—review and editing. C.M.: numerical modeling and data analysis. G.G. performed the numerical analysis on slope stability evaluation. P.W. revised the manuscript and summarized the work content. All authors have read and agreed to the published version of the manuscript.

**Funding:** This work was financially supported by the National Natural Science Foundation of China (52074020) and the Interdisciplinary Research Project for Young Teachers of USTB (Fundamental Research Funds for the Central Universities) (FRF-IDRY-21-001).

**Institutional Review Board Statement:** Not applicable.

**Informed Consent Statement:** Not applicable.

**Data Availability Statement:** The data will be available under request.

**Acknowledgments:** We thank Lan Zhou, Qin Ran, and Cui Yinxiang for their contibution in the experimental study of this work.

**Conflicts of Interest:** Author Kang Yuan was employed by the company Hainan Mining Co., Ltd. The remaining authors declare that the research was conducted in the absence of any commercial or financial relationships that could be constructed as a potential conflict of interest.

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
