# Peer review of "Slope Failure of Shilu Metal Mine Transition from Open-Pit to Underground Mining under Excavation Disturbance"

_applsci, doi:10.3390/app14031055_

Round 1

Reviewer 1 Report

Comments and Suggestions for Authors

This research paper presents an investigation of slope failure at the Shilu iron ore mine, Hainan Province, China. Detailed geological conditions, a numerical simulation, influencing factors were studied. In general, this manuscript shows an interesting case study. The paper can be after revision based on the following comments:

·         The main goal of this research should be pointed out in Abstract and Introduction

·         Lack of rock mass properties in Chapter 2

·         Lack of Open pit and Underground extraction description in Chapter 2 (slope angle, height of slope wall, height of a single UG extraction, UG extraction stages). Have OP and UG been extracted simultaneously for some period of time? slope angle, height of slope wall)

·         What is the justification of the rock mass properties given in chapter 3.1? Have they been calibrated whether has the numerical model been verified?

·         Lack of set of rock mass properties for modelling (both Mohr-Coulomb model and Mohr-Coulomb Strain softening model) in Chapter 3

·         Number of  word errors (highlighted in the PDF version) eg. transforming – transition, Iron mine – Iron ore mine, The crumbling method – the caving method, geotechnical soils – rocks, gravel – gobs, crumbling zone – caved zone. English should be improved.

·         Figures 5 and 6 are barely visible. Please improve their quality.

·         Title of chapter 2 and chapter 3 is the same. A mistake or it has to be in the given form?

·         Transition from OP to UG is an interesting aspect in Mining. Number of research have been studied. However, only a few references were cited in this manuscript. The more references cited the more better. Here are some suggested papers regarding OP-UG transition for reviewing. It may be useful for improving paper content and English:

-          Shi, S.; Guo, Z.; Ding, P.; Tao, Y.; Mao, H.; Jiao, Z. Failure Mechanism and Stability Control Technology of Slope during Open-Pit Combing Underground Extraction: A Case Study from Shanxi Province of China. Sustainability 2022, 14, 8939. https://doi.org/10.3390/su14148939.

-          Nguyen, P.M.V. Impact of longwall mining on slope stability – A case study. Studia Geotechnica et Mechanica, vol. 44, no.4, 2022, pp.282-295. https://doi.org/10.2478/sgem-2022-0019.

-          Hamman, ECF, Cowan, M, Venter, J & de Souza, JB 2020, 'Considerations for open pit to underground transition interaction', in PM Dight (ed.), Slope Stability 2020: Proceedings of the 2020 International Symposium on Slope Stability in Open Pit Mining and Civil Engineering, Australian Centre for Geomechanics, Perth, pp. 1123-1138, https://doi.org/10.36487/ACG_repo/2025_74

-          W.F. Visser, Optimization of the OP/UG Transition. Developmentof a Software Tool for Optimization of the Transition Depth andthe Open Pit Slope Angle – Main report. Technische Universiteit Delft (2006)

-          H. Peng, Q. Cai, W. Zhou, J. Shu, G. Li, Study on Stability ofSurface Mine Slope Influenced by Underground Mining belowthe Endwall Slope. Procedia Earth and Planetary Science. 2:7-13 (2011), https://doi.org/10.1016/j.proeps.2011.09.002

-          Cui, Y. X., Li, X. C., Jiang, Z. B., & Guo, F. F. (2012). Deformation Characteristics and Mechanism of the West Wall of Beiyi Iron Open-Pit Mine in Hainan Province. In Advanced Materials Research (Vols. 455–456, pp. 1394–1398). Trans Tech Publications, Ltd. https://doi.org/10.4028/www.scientific.net/amr.455-456.1394

-          D.A. Diaz, M.G. Schellman, Geomechanical status and actionplans for interaction between Andina subsidence craterand Los Bronces open pit, in PM Dight (ed.), Proceedings ofthe First Asia Pacific Slope Stability in Mining Conference,Australian Centre for Geomechanics, Perth. 613-628 (2016), https://doi.org/10.36487/ACG_rep/1604_41_Diaz.

-          BRUMMER R., LI H., MOSS A., The Transition from Open Pitto Underground Mining: an Unusual Slope Failure Mechanism at Palabora, Proceedings Int. Symposium on Stabilityof Rock Slopes, Cape Town 2006.

-          Joyce Chung, Mohammad Waqar Ali Asad, Erkan Topal, Timing of transition from open-pit to underground mining: A simultaneous optimisation model for open-pit and underground mine production schedules, Resources Policy, Volume 77, 2022, 102632, https://doi.org/10.1016/j.resourpol.2022.102632.

-          https://www.saimm.co.za/Conferences/RockSlopes/421-434_Olavarria.pdf

-          https://www.semanticscholar.org/paper/Study-on-the-Process-and-Mechanism-of-Slope-Failure-Hu-Ren/e57010aa3a4b5a14a3d0d32890bbb0cfeeebea92#citing-papers

-          Bakhtavar, E.; Shahriar, K.; Oraee, K. Transition from open-pit to underground as a new optimization challenge in mining engineering. J. Min. Sci. 2009, 45, 485–494.

Comments on the Quality of English Language

The English text should be double-check by native speakers for accuracy and fluency.

Author Response

Responds to the comments of Reviewer #1

This research paper presents an investigation of slope failure at the Shilu iron ore mine, Hainan Province, China. Detailed geological conditions, a numerical simulation, influencing factors were studied. In general, this manuscript shows an interesting case study. The paper can be after revision based on the following comments:

Thank you for the valuable comments. The replies are listed below.

  1. The main goal of this research should be pointed out in Abstract and Introduction.

Reply: Thank you for the suggestion. We have pointed out in Abstract and Introduction.

Abstract: “The research results can understand the stability of the open-pit to underground rock mass in Hainan, judge the development trend of surface subsidence range, and provide a reference for the stability evaluation of the rock mass mined by the open-pit to underground caving method.”

Introduction: “To describe the geomechanical phenomena that occur under the conditions of un-derground exploitation, this paper deals with the problem of open-pit slopes stability and ground surface under the influence of underground mining of the Shilu iron depos-it in China. Simulations with the help of numerical computational models, using the FLAC3D code were conducted to study the influence of staged exploitation of the ore body on the slope stability. The work may provide a methodology reference for the analysis of rock stability in open pit to underground mines.”

  1. Lack of rock mass properties in Chapter 2.

Reply: Thank you for the suggestion. These rock mass properties were introduced in the revised manuscript.

“In order to obtain accurate rock mechanical parameters, the physical and me-chanical indexes of each rock in the underground and mining areas were obtained through multiple processes such as on-site sampling, preparation of specimens, density test, wave velocity test, mechanical experiment, and data analysis, and the mechanical parameters of rock mass were determined by combining the Hoek-Brown empirical speculation method. The density of Shilu iron ore schist is about 2798.37 kg/m3, and the wave velocity is 5337.88 m/s. The density of the double permeable rock is about 2899.95 kg/m3, and the wave velocity is 5759.31 m/s. The hematite is about 3675.58 kg/m3 and the wave velocity is 5781.59 m/s. The average uniaxial compressive strength of schist in the North No. 1 mining area is 72.04 MPa, the elastic modulus is 29.55 GPa, the average secant modulus is 16.55 Gpa, and the Poisson's ratio is about 0.35. The uniaxial compressive strength of bipermeable rock and hematite is 68.09 MPa and 106.44 MPa, the elastic modulus is 36.97 MPa and 65.12 MPa, and the Poisson's ra-tio is 0.33. The average cohesion of the joint plane of the rock mass was 0.323 MPa, the average friction angle was 37.965°, and the average friction coefficient was 0.786. It provides important mechanical data for the subsequent mechanical analysis of engi-neering rock mass and the numerical simulation of mining subsidence.”

This information has been added to Part 2.2 of Chapter 2.

  1. Lack of Open pit and Underground extraction description in Chapter 2 (slope angle, height of slope wall, height of a single UG extraction, UG extraction stages). Have OP and UG been extracted simultaneously for some period of time? slope angle, height of slope wall)

Reply: Thank you for the comment. These Open pit and Underground extraction description have been added to Part 2.3 of Chapter 2 as follows:

The slope height of the open-pit mining of Shilu Iron Mine is 12m, the width of the slope design step is 6-10 m, the slope angle of the open-pit mine is 60-70°, and the final slope foot is 29°-49°, and it was transferred to underground mining in June 2018, with a single underground mining height of 15m, and the mining of ore bodies above -45 m has been completed.

  1. What is the justification of the rock mass properties given in chapter 3.1? Have they been calibrated whether has the numerical model been verified?

Reply: Thank you for the comment. The relevant justification for chapter 3.1 has been added, and the rock mass parameter characteristics are listed in Table 1.

Table 1 Mechanical parameters of rock mass

Modulus of elasticity

Poisson's ratio

Internal friction angle

Compressive strength

Tensile strength

Cohesion

65.12MPa

0.3

35°

106.44 MPa

8.87 MPa

25kPa

  1. Lack of set of rock mass properties for modelling (both Mohr-Coulomb model and Mohr-Coulomb Strain softening model) in Chapter 3.

Reply: Thank you for the comment. We have added rock mass properties for modelling in Table 2 in the revision.

Table 2 Model mechanical parameters

Lithology

Modulus of elasticity

Poisson's ratio

Internal friction angle

Compressive strength

Cohesion

Hematite

65.12MPa

0.33

57.91°

106.44 MPa

3.25kPa

Surrounding rock

36.97MPa

0.26

42.55°

68.09MPa

1.62kPa

  1. Number of word errors (highlighted in the PDF version) eg. transforming – transition, Iron mine – Iron ore mine, The crumbling method – the caving method, geotechnical soils – rocks, gravel – gobs, crumbling zone – caved zone. English should be improved.

Reply: Thank you for spotting. These error words are corrected accordingly in the revised manuscript. Moreover, we have also checked carefully and corrected some other errors.

  1. Figures 5 and 6 are barely visible. Please improve their quality.

Reply: Thank you for the suggestion. We have replaced Figures 5 and 6 with clearer images.

  1. Title of chapter 2 and chapter 3 is the same. A mistake or it has to be in the given form?

Reply: Thank you for spotting. The title of chapter 2 and chapter 3 are revised. The title of pt. 3 is revised to be “Numerical model”.

  1. Transition from OP to UG is an interesting aspect in Mining. Number of research have been studied. However, only a few references were cited in this manuscript. The more references cited the more better. Here are some suggested papers regarding OP-UG transition for reviewing. It may be useful for improving paper content and English:

-          Shi, S.; Guo, Z.; Ding, P.; Tao, Y.; Mao, H.; Jiao, Z. Failure Mechanism and Stability Control Technology of Slope during Open-Pit Combing Underground Extraction: A Case Study from Shanxi Province of China. Sustainability 2022, 14, 8939. https://doi.org/10.3390/su14148939.

-          Nguyen, P.M.V. Impact of longwall mining on slope stability – A case study. Studia Geotechnica et Mechanica, vol. 44, no.4, 2022, pp.282-295. https://doi.org/10.2478/sgem-2022-0019.

-          Hamman, ECF, Cowan, M, Venter, J & de Souza, JB 2020, 'Considerations for open pit to underground transition interaction', in PM Dight (ed.), Slope Stability 2020: Proceedings of the 2020 International Symposium on Slope Stability in Open Pit Mining and Civil Engineering, Australian Centre for Geomechanics, Perth, pp. 1123-1138, https://doi.org/10.36487/ACG_repo/2025_74

-          W.F. Visser, Optimization of the OP/UG Transition. Developmentof a Software Tool for Optimization of the Transition Depth andthe Open Pit Slope Angle – Main report. Technische Universiteit Delft (2006)

-          H. Peng, Q. Cai, W. Zhou, J. Shu, G. Li, Study on Stability ofSurface Mine Slope Influenced by Underground Mining belowthe Endwall Slope. Procedia Earth and Planetary Science. 2:7-13 (2011), https://doi.org/10.1016/j.proeps.2011.09.002

-          Cui, Y. X., Li, X. C., Jiang, Z. B., & Guo, F. F. (2012). Deformation Characteristics and Mechanism of the West Wall of Beiyi Iron Open-Pit Mine in Hainan Province. In Advanced Materials Research (Vols. 455–456, pp. 1394–1398). Trans Tech Publications, Ltd. https://doi.org/10.4028/www.scientific.net/amr.455-456.1394

-          D.A. Diaz, M.G. Schellman, Geomechanical status and actionplans for interaction between Andina subsidence craterand Los Bronces open pit, in PM Dight (ed.), Proceedings ofthe First Asia Pacific Slope Stability in Mining Conference,Australian Centre for Geomechanics, Perth. 613-628 (2016), https://doi.org/10.36487/ACG_rep/1604_41_Diaz.

-          BRUMMER R., LI H., MOSS A., The Transition from Open Pitto Underground Mining: an Unusual Slope Failure Mechanism at Palabora, Proceedings Int. Symposium on Stabilityof Rock Slopes, Cape Town 2006.

-          Joyce Chung, Mohammad Waqar Ali Asad, Erkan Topal, Timing of transition from open-pit to underground mining: A simultaneous optimisation model for open-pit and underground mine production schedules, Resources Policy, Volume 77, 2022, 102632, https://doi.org/10.1016/j.resourpol.2022.102632.

-          https://www.saimm.co.za/Conferences/RockSlopes/421-434_Olavarria.pdf

-          https://www.semanticscholar.org/paper/Study-on-the-Process-and-Mechanism-of-Slope-Failure-Hu-Ren/e57010aa3a4b5a14a3d0d32890bbb0cfeeebea92#citing-papers

-          Bakhtavar, E.; Shahriar, K.; Oraee, K. Transition from open-pit to underground as a new optimization challenge in mining engineering. J. Min. Sci. 2009, 45, 485–494.

Reply: Thank you for the suggestion. We have added the above references in the revised manuscript.

Reviewer 2 Report

Comments and Suggestions for Authors

The article deals with the problem of open-pit slopes stability and ground surface under the influence of underground mining of the Shilu iron deposit in China.

The analysis of these very complex geomechanical phenomena is very complicated and refers to surface crumble zones, displacement and subsidence zones, sikhole zone, produced as a result of the phased exploitation in depth of the iron ore deposit.

To describe the geomechanical phenomena that occur under the conditions of underground exploitation, the authors of the research developed in this article have resorted to a series of simulations with the help of numerical computational models, using the FLAC 3D finite difference software.

The simulations modeled the staged exploitation of the ore body, in depth, and the analysis highlighted the development of the state of stresses and deformations in the ground above the underground excavations, up to the surface, including the open-pit area, with the highlighting of areas of accentuated deformation and areas of collapse.

The description of deformation phenomena is scientifically argued in great detail.

The results of the numerical analysis were also validated with the help of data obtained from UAV photography technology.

The content of this article is very interesting, and through the geomechanical phenomena developed, it contributes to the enrichment of the scientific literature in this field, which is why it deserves to be disseminated through publication, but after correcting some minor language and drafting errors, such as:

-Lines 81 and 142: pt.2 and 3 contain the same title (the title from pt.3 must be changed);

-Line 164: kPa instead of KPa, etc.

Comments on the Quality of English Language

The article deals with the problem of open-pit slopes stability and ground surface under the influence of underground mining of the Shilu iron deposit in China.

The analysis of these very complex geomechanical phenomena is very complicated and refers to surface crumble zones, displacement and subsidence zones, sikhole zone, produced as a result of the phased exploitation in depth of the iron ore deposit.

To describe the geomechanical phenomena that occur under the conditions of underground exploitation, the authors of the research developed in this article have resorted to a series of simulations with the help of numerical computational models, using the FLAC 3D finite difference software.

The simulations modeled the staged exploitation of the ore body, in depth, and the analysis highlighted the development of the state of stresses and deformations in the ground above the underground excavations, up to the surface, including the open-pit area, with the highlighting of areas of accentuated deformation and areas of collapse.

The description of deformation phenomena is scientifically argued in great detail.

The results of the numerical analysis were also validated with the help of data obtained from UAV photography technology.

The content of this article is very interesting, and through the geomechanical phenomena developed, it contributes to the enrichment of the scientific literature in this field, which is why it deserves to be disseminated through publication, but after correcting some minor language and drafting errors, such as:

-Lines 81 and 142: pt.2 and 3 contain the same title (the title from pt.3 must be changed);

-Line 164: kPa instead of KPa, etc.

Author Response

Responds to the comments of Reviewer #2

The article deals with the problem of open-pit slopes stability and ground surface under the influence of underground mining of the Shilu iron deposit in China. The analysis of these very complex geomechanical phenomena is very complicated and refers to surface crumble zones, displacement and subsidence zones, sikhole zone, produced as a result of the phased exploitation in depth of the iron ore deposit. To describe the geomechanical phenomena that occur under the conditions of underground exploitation, the authors of the research developed in this article have resorted to a series of simulations with the help of numerical computational models, using the FLAC 3D finite difference software. The simulations modeled the staged exploitation of the ore body, in depth, and the analysis highlighted the development of the state of stresses and deformations in the ground above the underground excavations, up to the surface, including the open-pit area, with the highlighting of areas of accentuated deformation and areas of collapse. The description of deformation phenomena is scientifically argued in great detail. The results of the numerical analysis were also validated with the help of data obtained from UAV photography technology.

Reply: Thank you for the kind comments and encouragement. We have revised the manuscript carefully. The replies are listed below.

The content of this article is very interesting, and through the geomechanical phenomena developed, it contributes to the enrichment of the scientific literature in this field, which is why it deserves to be disseminated through publication, but after correcting some minor language and drafting errors, such as:

-Lines 81 and 142: pt.2 and 3 contain the same title (the title from pt.3 must be changed);

-Line 164: kPa instead of KPa, etc.

Reply: Thank you for spotting. The title of pt. 2 and pt. 3 are revised. The title of pt. 3 is revised to be “Numerical model”. The unit of kPa is corrected accordingly in the revised manuscript. Moreover, we have also checked carefully and corrected some errors.

Reviewer 3 Report

Comments and Suggestions for Authors

Dear authors,

After reading your paper entitled Slope Failure of Shilu Metal Mine Transforming from Open-pit to Underground Mining under Excavation Disturbance I conclude the following:

- the paper is well structured and the research results presented in it are interesting;

- I have no observations regarding the research part, but only two observations on the content, namely: at R108-109 you say mechanical parameters are shown in the table? this table is missing; also the part of references should be internationalized (it is mostly national).

Author Response

Responds to the comments of Reviewer #3

Dear authors, After reading your paper entitled Slope Failure of Shilu Metal Mine Transforming from Open-pit to Underground Mining under Excavation Disturbance I conclude the following: - the paper is well structured and the research results presented in it are interesting; I have no observations regarding the research part, but only two observations on the content, namely: at R108-109 you say mechanical parameters are shown in the table? this table is missing; also the part of references should be internationalized (it is mostly national).

Reply: Thank you for spotting. As for the table, we have added the mechanical parameters and listed in Table 1 in the revision. As for the references, we have added more international references.